# The association between lead exposure and crime: A systematic review

**Maria Jose Talayero**[1☯]*, **C. Rebecca Robbins**[1☯], **Emily R. Smith**[2‡], **Carlos Santos-Burgoa**[2‡]

**1** Department of Environmental and Occupational Health, The George Washington University, Washington, DC, United States of America, **2** Department of Global Health, The George Washington University, Washington, DC, United States of America

☯ These authors contributed equally to this work.
‡ ERS and CSB also contributed equally to this work.
* mjtalayero@gwu.edu

**Data Availability Statement:** This is a systematic review, the search strategy is included in the appendix, and all results are reported in the manuscript.

## Abstract

Prior research has demonstrated an association between lead exposure and criminal behavior at the population-level, however studies exploring the effect of lead exposure on criminal behavior at the individual-level have not been reviewed systematically. The intent of this study is to complete a systematic review of all studies assessing individual-level exposures to lead and the outcomes of crime and antisocial behavior traits. We included peer reviewed studies that were published prior to August 2022 and were classified as cohort, cross-sectional, or case-control. Studies measuring the outcomes of crime, delinquency, violence, or aggression were included. The following databases were searched using a standardized search strategy: ProQuest Environmental Science Database, PubMed, Tox-Net and the Public Affairs Information Service (PAIS). Seventeen manuscripts met our inclusion criteria. Blood lead was measured in 12 studies, bone lead in 3 studies, and dentine lead levels in 2 studies. This systematic review identified a wide range of diverse outcomes between exposure to lead at multiple windows of development and later delinquent, criminal and antisocial behavior. A review of all potential confounding variables included within each study was made, with inclusion of relevant confounders into the risk of bias tool. There is limited data at the individual level on the effects of prenatal, childhood, and adolescent lead exposure and later criminal behavior and more evidence is necessary to evaluate the magnitude of the associations seen in this review. Our review, in conjunction with the available biological evidence, suggests that an excess risk for criminal behavior in adulthood exists when an individual is exposed to lead in utero or in the early years of childhood. The authors report no conflict of interest and no funding source.

**Clinical trial registration:** PROSPERO ID: CRD42021268379.

## Introduction

Individuals that consistently experience high levels of lead exposure suffer from a variety of negative health impacts, including impairment of the renal and cardiovascular systems,

**Funding:** The author(s) received no specific funding for this work.

**Competing interests:** The authors have declared that no competing interests exist.

reproductive toxicity, immune system dysfunction, and delayed growth [1–3]. Damage to the nervous system from lead exposure can result in a variety of neurological effects, including a reduction in overall cognitive function, negative behavioral changes, lowered intelligence quotient (IQ) scores, decreased learning ability, poor memory scores, and impaired comprehension and reading abilities [4]. The impacts of lead exposure on the pediatric population are particularly severe, as children demonstrate a consistently higher physiological uptake of lead than adults, predisposing them to irreversible neurological impacts [2, 3]. The Centers for Disease Control have stated that there is "no safe level of lead exposure for children", and recently lowered the blood lead reference value from 5 μg/dL to 3.5 μg/dL [1, 5]. Sources of exposure to lead vary by country: in low- and middle-income (LMIC) populations, typical routes of lead exposure include pollutants from industrial waste, paint, glazed-clay pottery, traditional medical treatments (such as Daw Tway, a digestive aid), batteries, and various food sources [6]. In high-income countries (HIC), lead exposure is more likely to occur from industrial exposures, paints, and products that are imported into the country, such as children's toys and ceramics [6]. Although the dangers of lead exposure are well documented, lead exposure remains an issue of concern for both high- and low and middle-income countries [6].

Time-series ecological studies have previously explored the relationship between lead exposure changes and criminal behavior. These studies demonstrated positive associations between air lead concentrations, crime rates, and homicide rates at the aggregate level [7–9]. Animal studies have demonstrated impairments in memory, attention, spatial recognition, sensory function, and overall learning ability in those animals exposed to high levels of lead both in utero and in infancy, findings which translated to an impairment of neurobehavioral function in adulthood [10, 11]. There are a litany of molecular mechanisms that have been proposed as potential modalities of lead-mediated neurotoxicity. Major theories include inhibition of claudin-1, a protein that assists in maintaining a patent blood brain barrier in childhood; competition with divalent cations such as calcium which then inhibit normal synaptic function; mitochondrial dysfunction via decreased levels of glutathione; the creation of reactive oxidative stress species; and neural inflammation via microglial and astroglial inflammatory mechanisms [12]. For a deeper discussion on the potential molecular mechanisms of lead exposure on the neurological system, the reader is referred to Virgolini and Aschner's Molecular Mechanisms of Lead Neurotoxicity. Despite this, epidemiologic studies attempting to determine in individual subjects the relationship between lead exposure and crime have reported inconsistent findings, potentially identifying an ecologic fallacy in which aggregate data inaccurately represent findings at the individual level [13]. This review aims to systematize and determine the strength, magnitude, and consistency of the available evidence in studies assessing the association between an individual's lead exposure and the subsequent crime incidence.

## Methods

We registered the protocol for this meta-analysis via PROSPERO (ID: CRD42021268379) on August 19, 2021, and the results of the review were reported in accordance with Preferred Reporting Items for Systematic reviews and Meta-Analyses (PRISMA) guidelines [14].

### Ethics statement

An ethics statement is not applicable because this systematic review is based exclusively on published literature. Ethics and IRB approval was not required due to this reliance on published literature. Anonymity of all subjects is ensured due to the design of the review.

### Study question and eligibility criteria

The aim of our study was to address the research question, "What is the association between lead exposure and criminal behavior?" We included all studies with individual-level data and outcomes related to criminal behavior such as delinquency and aggressive behavior. Studies that included the following outcomes were included: crime, delinquency, violence, and aggressiveness. Additionally, we included studies that were published prior to August 2022, peer-reviewed, and were classified as cohort, cross-sectional, or case-control studies. All ecological and animal studies were excluded. All studies reporting on other negative behavioral outcomes associated with lead exposure were excluded.

### Information sources & search strategy

A librarian from The George Washington University developed the search strategy. The following databases were used for our search: ProQuest Environmental Science Database, PubMed, Toxnet and the Public Affairs Information Service (PAIS), using the following terms: "Lead" "Blood lead" OR "Lead Exposure", "Lead Exposure, Nervous System", "Lead Toxicity", "Blood lead levels", "Lead Poisoning", "Crime", "Criminal behavior", "Convictions", "Arrests", "Delinquency", "Violent", "Violence", "Aggressiveness", "Violent Behavior", and "Aggressive Behavior". (The complete search strategy can be found in S1 Text).

### Selection process

All identified studies were imported into Covidence, a systematic review tool. We removed all duplicate studies, and the remainder of the studies were assessed by two independent reviewers (MJT & CRR) to verify that inclusion and exclusion criteria were met. All studies were first screened using an abstract screening process intended to rule out ineligible studies, and the studies remaining were then assessed in depth to verify that all inclusion and exclusion criteria were met. When the reviewers did not agree, a third independent reviewer was consulted regarding the exclusion or inclusion of said study (SB & ES).

### Data collection process

Two reviewers individually and independently extracted data from each study into a spreadsheet. Both tables were then compared, and discrepancies were discussed and resolved among the reviewers. When results or methods were not clear, reviewers contacted the study authors for clarification.

### Data items and effect measures

From each of the studies, we extracted the number of participants, age at biomarker sample, mean (standard deviation) for each biomarker sample, outcome, outcome ascertainment methodology (i.e., bone, blood, dentine, etc.), and results. Outcomes were stratified by exposure time. All results are reported as odds ratios (OR), risk ratios (RR), incidence rate ratios (IRR) or beta coefficients.

### Risk of bias

A modified ROBINS-E tool was utilized to assess the risk of bias in each observational study of exposure (S1 and S2 Tables) [15]. Confounding bias, selection bias, exposure misclassification, missing data bias, outcome measurement bias, reporting bias, and overall bias were all assessed and ranked on a scale of low to very high and scored accordingly. Two independent reviewers

completed risk of bias reviews, and then conferred to reach consensus on all domains, with a third party adjudicating any conflicts that remained unresolved.

## Synthesis methods

We originally intended to pool data from studies with a common exposure and outcome metric, however there were only 2 studies with such characteristics, therefore a quantitative meta-analysis could not be completed. We thus conducted a narrative synthesis of the available studies and grouped the results by age at lead exposure assessment, with age categories as follows: prenatal, early childhood (≤6 years of age), late childhood (>6 years of age), and adolescence and adulthood (13 years of age and above).

## Results

### Study selection

As demonstrated by the PRISMA flow chart (Fig 1), 65 full-text studies were assessed for eligibility, with 50 studies excluded due to inappropriate exposure classification, outcome classification, or population studied. (Fig 1). There were six notable ecological studies that were excluded: *The Relationship Between Lead Exposure and Homicide* by Stretesky & Lynch (2001) [9], *The Relationship Between Lead and Crime* by Stretesky and Lynch (2004) [16], *Aggregate-level lead exposure, gun violence, homicide, and rape* by Boutwell et al., (2017) [17], *How Lead Exposure Relates to Temporal Changes in IQ, Violent Crime, and Unwed Pregnancy* by Nevin (2000) [7], *Lead Exposure and Behavior*: *Effects on Antisocial and Risky Behavior Among Children and Adolescents* by Reyes (2015) [18], and *Understanding international crime trends*: *The legacy of preschool lead exposure* by Nevin (2007) [8].

Included in the final review are 17 manuscripts representing 13 studies. Of the 13 studies, there were eight observational cohort studies that were utilized in 14 manuscripts included in this review: The Cincinnati Lead Study [19–21], The Birth-to-20 Plus Study [22–24], the Project on Human Development in Chicago Neighborhoods [25], The Pittsburgh Youth Study [26], the New Zealand Dunedin Multidisciplinary Health and Development Study [27], the Christchurch Health and Development Study [28], the Rhode Island Lead and Juvenile Delinquency Study [29], the Edinburgh Lead Study [30], and a cohort selected from the Milwaukee, Wisconsin DataShare database [31]. The Cincinnati Lead Study cohort and the Birth-to-20-Plus cohort were utilized multiple times by authors to create a total of 6 manuscripts, while the Pittsburgh Youth Study was utilized twice to create a total of 2 manuscripts (1 case-control and 1 observational cohort).

Wright et al., utilized data from participants within The Cincinnati Lead Study, a birth cohort that was recruited from 1979–1984. In their 2008 study, the authors measured average exposures of 250 participants within the study from three timepoints: prenatal exposure, average childhood exposure (assessed as an average of the blood lead levels collected over months 3–78) and late childhood exposure (measured at 6.5 years). Outcome was assessed as the average number of criminal arrests that occurred after the age of 18. The focus of the 2008 study was the association between blood lead levels throughout the entirety of the preschool and early school years and total official arrests in adulthood [20].

In his 2021 study, Wright et al., again utilized data from The Cincinnati Lead Study, using the same population and same time points of exposure from his 2008 study, but this time opting to subdivide the outcome into arrests from the ages of 18–24 and arrests from the ages of 27–33. This allowed the study authors to re-assess their original study findings by further extending the window of time in which a participant had been in adulthood by 10 years to see if the findings changed over time. Additionally, the 2021 study differentiated the type of arrest

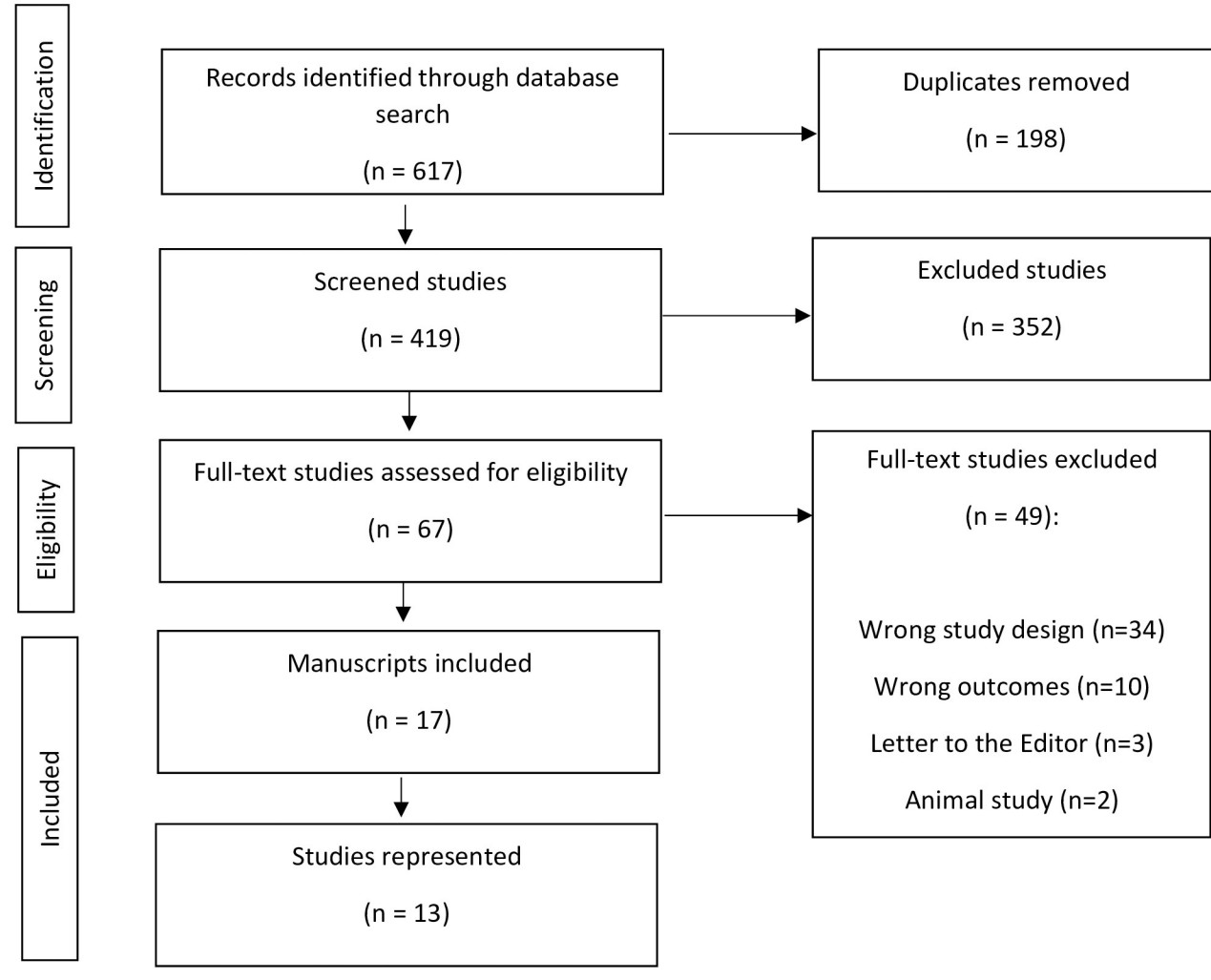

**Fig 1. PRISMA flow diagram.**

(i.e., total adult arrests, drug arrests, violent arrests, property arrests) and then controlled for all arrests that occurred prior to 2003 in order to measure the association between blood lead levels and total adult arrests later in adulthood (corresponding to study participants ages 27–33). For these reasons, we decided that both studies by Wright should be included in our systematic review, despite use of the same cohort and same participants [21].

Dietrich et al. utilized 195 participants from The Cincinnati Lead Study, but in contrast to the Wright et al., 2008 and 2021 studies, chose to consider only juvenile delinquency and anti-social behavior as measured by self and parental report surveys administered when the participants were between the ages of 15–17 [19].

Three manuscripts from South Africa utilized the longitudinal Birth-to-20 Plus cohort (BT20+), consisting of 3,273 participants born to mothers within public health facilities in Soweto and Johannesburg between April 23 to June 8, 1990 [22–24]. In their 2017 manuscript, Nkomo et al., 2017, analyzed participants with blood lead levels at age 13 who completed the Youth Self-Report survey at ages 14–15 in order to assess violent behavior in adolescence.

In contrast, Nkomo et al., 2018 analyzed the same set of participants as the 2017 study using the same blood lead level measure at age 13, but this time aimed to assess disobedience, direct

aggression, and indirect aggression in adolescence, again using the Youth Self-Report survey [22, 23].

A third manuscript was published from a sub-cohort from the Birth-to-20 Plus cohort, measuring bone lead levels in 100 young adults (23–24 years). Aggression was assessed using the Buss-Perry Aggression Questionnaire [24]. We decided to include both Nkomo et al., and Tlotleng et al., due to the clear differentiation between violent behavior and feelings of aggression in the outcomes assessed, as well as the inclusion of bone lead measurements.

## Study characteristics

Two studies originated from New Zealand [27, 28], 4 from South Africa [22–24, 32], 1 from Brazil [33], 1 from Scotland [30], 1 from Italy [34], and the remaining 9 from the United States (Tables 1 and 2) [19–21, 25, 26, 29, 31].

Only 1 study was a case-control study, conducted by Needleman et al., in Allegheny County, Pennsylvania, USA [26]. There were 4 cross-sectional studies [30, 32–34], with the remaining 12 studies classified as cohort studies.

## Study characteristics: Exposure status

Exposure status was primarily assessed via blood lead levels, while 3 studies measured bone lead levels via x-ray fluorescence spectroscopy [24, 26], and 2 measured dentine lead levels [28, 33]. Three studies measured prenatal blood lead levels [19–21], 6 measured blood lead levels from ages 0–6 [19–21, 25, 29, 31], 3 measured bone and blood lead levels from ages 6–11 [28, 30, 34], 4 measured blood and bone lead levels in adolescence [22, 23, 33, 36], and 2 measured adult blood and bone lead levels [24, 32]. Blood lead levels represent acute exposures, as the half-life of lead is roughly 36 days [35]. Despite this, exposure to lead is typically monitored via blood lead levels. Bone and dentine lead levels (half-life of years to decades) [36] are a better matrix from which to determine cumulative exposure levels [35].

**Table 1. Study design, year, and location of study.**

| Study | Manuscript | Year | Type | Location |
|---|---|---|---|---|
| The Edinburgh Lead Study | Thomson et al., | 1989 | Cross-Sectional | Edinburgh, Scotland |
| The Pittsburgh Youth Study | Needleman et al., | 1996 | Cohort | Pittsburgh, PA, US |
| The Cincinnati Lead Study | Dietrich et al., | 2001 | Cohort | Cincinnati, OH, US |
| The Pittsburgh Youth Study | Needleman et al., | 2002 | Case-Control | Pittsburgh, PA, US |
| The Christchurch Health and Development Study | Fergusson et al., | 2008 | Cohort | Christchurch, New Zealand |
| The Cincinnati Lead Study | Wright et al., | 2008 | Cohort | Cincinnati, OH, US |
| The Brazil Study | Olympio et al., | 2010 | Cross-Sectional | Sao Paulo, Brazil |
| New Zealand Dunedin Multidisciplinary Health and Development Study | Beckley et al., | 2018 | Cohort | South Island, New Zealand |
| The Birth-to-20 Plus cohort | Nkomo et al., | 2017 | Cohort | Soweto-Johannesburg, South Africa |
| Lead Exposure in Shooters Study | Naicker et al., | 2018 | Cross-Sectional | Gauteng, South Africa |
| Project on Human Development in Chicago Neighborhoods | Sampson and Winter | 2018 | Cohort | Chicago, IL, US |
| The Birth-to-20 Plus cohort | Nkomo et al., | 2018 | Cohort | Soweto-Johannesburg, South Africa |
| Rhode Island Lead and Juvenile Delinquency Study | Aizer and Currie | 2019 | Cohort | Rhode Island, US |
| Wisconsin DataShare database | Emer et al., | 2020 | Cohort | Milwaukee, WI, US |
| The Cincinnati Lead Study | Wright et al., | 2021 | Cohort | Cincinnati, OH, US |
| The Neurotoxic Elements on Italian Schoolchildren Study | Renzetti et al., | 2021 | Cross Sectional | Taranto, Italy |
| The Birth-to-20 Plus cohort Sub cohort: Bone Health Cohort | Tlotleng et al., | 2022 | Sub-Cohort | Johannesburg, South Africa. |

**Table 2. Characteristics of included studies.**

| Characteristics of Included Studies | |
|---|---|
| **Design** | |
| Cohort | 12 (71%) |
| Case control | 1 (6%) |
| Cross Sectional | 4(23%) |
| **Location** | |
| United States | 9 (50%) |
| Europe | 2 (11%) |
| Africa | 4 (22%) |
| South America | 1 (6%) |
| New Zealand | 2 (11%) |
| **Exposure Measurement** | |
| Blood Lead Levels | 12 (70%) |
| Venous | 8 (67%) |
| Capillary | 1 (8%) |
| Both | 3 (25%) |
| Bone lead levels | 3 (18%) |
| Dentine lead levels | 2 (12%) |
| Dental: enamel | 1 (50%) |
| Deciduous teeth | 1 (50%) |
| **Outcome Measurement** | |
| Official Reports | 5 (29%) |
| Official + Self or Parental | 3 (18%) |
| Self and Parental Report | 2 (12%) |
| Parental/Caregiver Report | 3 (18%) |
| Self-Report | 4 (23%) |

## Study characteristics: Outcome measures

A wide variety of outcome measures were utilized by the study authors to define "criminal behavior," with some authors choosing to report primarily on violent arrests, some choosing to focus on aggressive behavior irrespective of arrest status, and some combining non-violent arrests (drug arrests and property crime arrests) into their violent crime outcomes. Violent crime arrests and property crime arrests were the most commonly reported outcomes across studies [20, 21, 26, 27, 29], although these outcomes were frequently reported in conjunction with additional outcomes such as lifetime arrests. Table 3 displays the details of the various outcome ascertainment strategies used by each study author.

Self-report of aggression, delinquency, or crime was a common outcome assessment modality, with a total of 7 studies utilizing a variety of both questionnaires and interviews to assess criminal behavior. Four studies used self-report as their sole outcome measure [19, 26, 30, 33]. The questionnaires that was used across more than one study were the Youth Self-Report Questionnaire [37], utilized by Nkomo et al., in her 2017 and 2018 studies [22, 23], and the Buss-Perry Questionnaire [38], utilized by Naicker et al., [32] and Tlotleng et al., [24] Only two authors opted to utilize one-on-one interviews [27, 28], and both assessed their respective outcomes in conjunction with official criminal records to optimize accuracy. The remaining two questionnaires used were the Self and Parental Report of Delinquent Behavior [39, 40], utilized by Dietrich et al., [19]. The remaining studies used official court records and official police arrest counts to assess their outcomes (Table 3).

**Table 3. Summary of the link between lead exposure and crime and violence, stratified by age of exposure assessment and study.**

| Author, Year Site (Study design) | N | Age at Exposure Measures | Exposure Metric per Participant | Mean (SD) (μg/dL) / (μg/g) | Age at Outcome Measures | Outcomes and Results | Overall Bias |
|---|---|---|---|---|---|---|---|
| **Prenatal Pb Exposure** | | | | | | | |
| Dietrich, K, N., et al., 2001 Cincinnati (Cohort) | 157 | Prenatal | One maternal venous BPb | 8.9 (3.9) | 15–17 y | B coefficient (SE) p-value Self-Report: 0.192 (0.76) p = 0.002* Parental Report: 0.194 (0.089) p = 0.032* | Moderate |
| Wright, J. P., et al., 2008 and 2021 Cincinnati (Cohort) | 254 | Prenatal | One maternal venous BPb | 8.3 (3.8) | 18–33 y | IRR(95% CI) Adult arrests: 1.15 (1.03, 1.27)* Violent arrests:1.17 (0.98, 1.40) Drug arrests: 1.21 (1.02, 1.43)* Property arrests:0.96 (0.80, 1.14) Lifetime arrests: 1.16 (1.06, 1.28)* | Low |
| **Early Childhood Pb Exposure ≤6y** | | | | | | | |
| Sampson, R. & Winter, A., 2018 Chicago (Cohort) | 212 | 0 - < 6 y | Average childhood capillary or venous BPb | 6.2 (4.6) | 16–21 y | OLS (Coefficients and 95%CI) Lead and antisocial behavior: 0.22 (0.00, 0.45) Lead and Arrests: -0.06 (-0.16, 0.03) Lead and violent arrests: 0.02 (-0.07, 0.11) Antisocial behavior and arrests: 0.56 (0.18, 0.94)* Antisocial behavior and violent arrests: 0.55 (0.08, 1.03)* | Low |
| Aizer, A. & Currie, J., 2019 Rhode Island (Cohort) | 124,579 | 0 - < 6 y | BPb childhood geometric capillary or venous mean | 3.8 (4.8) | 15–23 y | Juvenile or adult detentions or incarceration OLS (Point estimates and Robust SE) 0.0014 (0.0002)* | Low |
| Emer, L. R., et al., 2020 Milwaukee (Cohort) | 89,129 | 0 - < 6 y | Average childhood and Peak capillary or venous BPb | Median (IQR) 5.5 (5.5) 7.0 (8.0) | > 12 y | Firearm violence perpetration RR (95% CI) Peak ≥5 < 10 μg/dL: 2.5 (1.5, 4.1)* Peak ≥10 < 20 μg/dL: 3.1 (1.9, 5.2)* Peak ≥20 μg/dL: 3.5 (2.1, 5.8)* | Moderate |
| Wright, J. P., et al., 2008 and 2021 Cincinnati (Cohort) | 254 | 1 m– 6 y | Average childhood venous BPb | 14.4 (6.6) | 18–33 y | IRR (95% CI) Adult arrests: 1.03 (0.99, 1.08) Violent arrests: 1.01 (0.94, 1.09) Drug arrests: 1.06 (0.98, 1.15) Property arrests: 1.03 (0.96, 1.11) Lifetime Arrests: 1.02 (0.97, 1.07) | Low |
| Dietrich K. N., et al., 2001 Cincinnati (Cohort) | 186 | 3 m– 6 y | Average childhood venous BPb | <10 μg/dL: n 63 10–15 μg/dL: n 63 16–20 μg/dL: n 43 >20 μg/dL: n 26 | 15–17 y | B coefficient (SE) p-value Self-Report Delinquent Behavior: 0.101 (0.47) p = 0.36 Parental Report Delinquent Behavior: 0.090 (0.56) p = 0.109 | Moderate |

*(Continued)*

**Table 3.** (Continued)

| Author, Year Site (Study design) | N | Age at Exposure Measures | Exposure Metric per Participant | Mean (SD) (µg/dL) / (µg/g) | Age at Outcome Measures | Outcomes and Results | Overall Bias |
|---|---|---|---|---|---|---|---|
| Wright, J. P., et al., 2008 and 2021 Cincinnati (Cohort) | 254 | 5–6.5 y ** | Average late childhood venous BPb | n.a. | 18–33 y | IRR (95% CI) Adult arrests: 1.07 (1.01, 1.13)* Violent arrests: 1.04 (0.96, 1.14) Drug arrests: 1.13 (1.03, 1.24)* Property arrests: 1.02 (0.93, 1.12) Lifetime arrests 1.06 (1.00,1.13) * | Low |
| Wright, J. P., et al., 2008 and 2021 Cincinnati (Cohort) | 254 | 6 y | One late childhood venous BPb | 8.3 (4.8) | 18–33 y | IRR (95% CI) Adult arrests: 1.07 (1,1.14)* Violent arrests: 1.08 (0.96, 1.21) Drug arrests: 1.17 (1.02, 1.33)* Property arrests: 1 (0.89, 1.12) Lifetime arrests: 1.08 (1.01, 1.16)* | Low |
| **Late Childhood Pb Exposure >6y - <13y** | | | | | | | |
| Dietrich, K,N. et al., 2001 Cincinnati (Cohort) | 186 | 6.5 y | One late childhood | <10 µg/dL: n 63 10–15 µg/dL: n 63 16–20 µg/dL: n 43 >20 µg/dL: n 26 | 15–17 y | B coefficient (SE) p-value Self-Report Delinquent Behavior: 0.193 (0.61) p = 0.002* Parental Report Delinquent Behavior: 0.131 (0.072) p = 0.070 | Moderate |
| Fergusson, D.M., et al., 2008 New Zealand (Cohort) | 871 | 6–9 y | One childhood dentine Pb from deciduous teeth | 6.2 µg/g | 14–21 y | Violent/property convictions and offenses Negative Binomial Regression (Coefficients, SE, and p value) 6–8 µg/g: 0.35 (0.18) p = 0.02* 9–11 µg/g: 0.52 (0.18) p = 0.02* >12 µg/g: 0.79 (0.18) p = 0.02* | Moderate |
| Thomson, G.O.B., et al., 1989 Edinburgh (Cross-Sectional) | 501 | 6–9 y | One childhood venous BPb | 10.4 | 6–9 y | Parent and teacher report of aggressive/anti-social behavior Log-odds coefficient Log blood-lead increase: 1.08 (p = .004)* | Serious |
| Renzetti, S., et al., 2021 Italy (Cross-Sectional) | 299 | 6–11 y | One childhood venous BPb | 0.94 (0.48) | 6–11 y | Tobit regression Tobit marginal effect coefficients (95% CI) p value Rule breaking behavior: 1.3 (−0.3, 2.9) p = >0.05 Aggressive Behavior: 2.2 (0.5, 4.0) p = <0.05* | Low |
| Needleman, H.L., et al., 1996 Pittsburgh (Cohort) | 212 | 9–14 y | Two bone lead measures (tibial) | n.a. | 9–14 y | Parent and teacher report of delinquency OR (95% CI) Parent: 1.89 (0.83, 4.3) Teacher: 2.16 (0.96, 4.6) | Moderate |

(Continued)

**Table 3.** (Continued)

| Author, Year Site (Study design) | N | Age at Exposure Measures | Exposure Metric per Participant | Mean (SD) (µg/dL) / (µg/g) | Age at Outcome Measures | Outcomes and Results | Overall Bias |
|---|---|---|---|---|---|---|---|
| Beckley, A.L., et al., 2018 New Zealand (Cohort) | 553 | 11 y | One late childhood venous BPb | 11.0 (4.6) | 15, 18, 21, 26, 32, and 38 y | OR (95%CI) Criminal convictions: 1.23 (1.00, 1.51)* One time offense: 1.25 (0.95, 1.64) Recidivistic offender: 1.21 (0.93, 1.57) Nonviolent offender: 1.28(1.01, 1.61)* Violent offender 1.13 (0.82, 1.55) | Moderate |
| **Adolescence and Adult Pb Exposure >13y** | | | | | | | |
| Nkomo, P., et al., 2017 South Africa (Cohort) | 1332 | 13 y | One early adolescence venous BPb | 5.8 (2.4) | 15–16 y | B coefficient (SE) p value Violence using a weapon: -0.07 (0.01) p>0.05 Physical violence: 0.26 (0.01) p <0.0001* Fighting: 0.08 (0.07) p>0.05 Sexual harassment: 0.01(0.07) p>0.05 Robbing: -0.03 (0.07) p>0.05 Verbal and emotional abusive behavior: -0.05 (0.08) p>0.05 | Low |
| Nkomo, P., et al., 2018 South Africa (Cohort) | 1086 | 13 y | One early adolescence venous BPb | 5.6 (2.3) | 14–15 y | B coefficient (SE) p value 5–9.99 µg/dl Indirect aggression: 0.01 (0.06) p = 0.94 Direct Aggression: -0.07 (0.06) p = 0.28 Disobedience:0.10 (0.07) p = 0.14 >10 µg/dl Indirect aggression:0.26 (0.18) p = 0.16 Direct Aggression: 0.34 (0.06) p = 0.02* Disobedience:0.28 (0.20) p = 0.17 | Low |
| Olympio, K., et al., 2010 Brazil (Cross-Sectional) | 173 | 14 – 18y | Dental enamel lead levels | Clinical 197.1 (157) Normal 177.5 (347.0) | 14–18 y | OR (95%CI) Aggressive behavior: 1.31 (0.42,4.09) Rule-breaking behavior: 3.72 (0.99, 14.04) Self-Reported delinquency: 0.93 (0.40, 2.20) | Serious |
| Needleman, H.L., et al., 2002 Pittsburgh (Case-Control) | 195 Cases / 150 Controls | 15–17 y | Bone lead levels (tibial) | Cases 11 (32.7 µg/g) Controls 1.5 (32.1 µg/g) | 12–18 y | Arrest or adjudication as delinquent by juvenile court OR (95%CI) 3.7 (1.3, 10.5)* | Moderate |
| Naicker, N., et al., 2018 South Africa (Cross-Sectional) | 87 | 18–74 y | One adulthood capillary BPb | 11.9 | 18–74 y | OR (95%CI) Hostility: ≥10 µg/dL: 2.83 (1.103, 7.261)* | Serious |

(Continued)

**Table 3.** (Continued)

| Author, Year Site (Study design) | N | Age at Exposure Measures | Exposure Metric per Participant | Mean (SD) (µg/dL) / (µg/ g) | Age at Outcome Measures | Outcomes and Results | Overall Bias |
|---|---|---|---|---|---|---|---|
| Tlotleng, N., et al., 2022 South Africa (Cohort) | 100 | 9 y | Bone lead levels (tibial) | 8.7 (5.3) (µg/g) | 24–25 y | Coefficients (95% CI) p value Anger 0.25 (0.04–0.37) p = 0.017* Physical Aggression 0.093 (−0.01–0.27) p = 0.35 Verbal Aggression 0.093 (−0.05–0.23) p = 0.18 Hostility 0.030 (−0.19–0.26) p = 0.79 | Low |

n.a. = Not available

* Significant results

** Study assessed 5–6.5y.o, kept in the early childhood category for classification purposes.

## Participant characteristics

Participant characteristics across included studies varied widely. Fourteen of the included studies had less than 1000 study subjects (median = 300), with 2 including a population ≥89,000 subjects. Those participants from the Cincinnati Lead Study, the Milwaukee, and Wisconsin (USA) cohort were predominantly African American and tended towards higher blood lead levels and lower socioeconomic status at baseline. Olympio et al., [33] drew their study sample from adolescents residing in the slums of Bauru, Brazil, an area notorious for high crime and low socioeconomic status. Conversely, cohorts such as The Dunedin Multidisciplinary Health and Development cohort and the Rhode Island cohort represented a large range of blood lead levels, races/ethnicities, and socioeconomic statuses [27, 29] (Table 3).

## Risk of bias

Risk of bias was assessed using a modified version of the final Risk of Bias in Non-Randomized Studies of Exposure (ROBINS-E) tool (S1 Table). Risk of bias across all studies was generally low to moderate, with only 3 studies categorized as "very high" risk of bias: Naicker et al., Olympio et al., and Thomson et al., [30, 32, 33]. Nine of the 15 studies suffered from some level of confounding bias, 11 from some degree of exposure classification bias, and 8 from some degree of missing data bias. Conversely, there was little reporting bias and selection bias across studies (Table 4).

## Results of synthesis

**Prenatal exposure.** Two studies assessed the relationship between prenatal lead exposure and crime, and both studies used data from *The Cincinnati Lead Study* cohort [19–21]. Dietrich et al., used self and parental reports of delinquent behavior at 15–17 years of age and reported a statistically significant beta coefficient for self-reported delinquent behavior (β = .192, p < .001) 0.192 (0.76), as well as a statistically significant beta coefficient for parental reported delinquent behavior (β = .194, p = .032) 0.194 (0.089). Wright et al., used official arrest data and found an increased risk between adult arrests ((IRR 1.15 (1.03–1.27)), drug arrests ((IRR 1.21 (1.02–1.43)), lifetime arrests ((IRR 1.16 (1.06–1.28) and prenatal lead exposure, however no association was seen with violent arrests (IRR 1.17(0.98, 1.40)), or property arrests (IRR 1 (0.89, 1.12)).

**Table 4. Risk of bias assessment ROB assessment.**

| Study Author | Confounding Bias | Selection Bias * | Exposure Classification | Missing Data Bias | Outcome Measurement | Reporting Bias | Overall Bias |
|---|---|---|---|---|---|---|---|
| Aizer & Currie, 2019 | Low | Low | Moderate | Low | Moderate | Low | Low |
| Beckley et al., 2018 | Serious | Prospective cohort study | Low | Moderate | Moderate | Low | Moderate |
| Dietrich et al., 2001 | Low | Prospective cohort study | Low | Moderate | Moderate | Low | Moderate |
| Emer et al., 2020 | Serious | Low | Moderate | Moderate | Serious | Low | Moderate |
| Fergusson et al., 2008 | Low | Prospective cohort study | Moderate | Low | Moderate | Low | Low |
| Naicker et al., 2018 | Moderate | Serious | Moderate | Critical | Critical | Low | Serious |
| Needleman et al., 1996 | Moderate | Prospective cohort study | Moderate | Critical | Moderate | Low | Moderate |
| Needleman et al., 2002 | Serious | Low | Moderate | Moderate | Moderate | Low | Moderate |
| Nkomo et al., 2017 | Low | Prospective cohort study | Moderate | Low | Low | Low | Low |
| Nkomo et al., 2018 | Low | Prospective cohort study | Moderate | Low | Low | Low | Low |
| Olympio et al., 2010 | Moderate | Serious | Moderate | Moderate | Serious | Low | Moderate |
| Renzetti et al., 2021 | Low | Low | Low | Low | Low | Low | Low |
| Sampson & Winter, 2018 | Low | Low | Low | Low | Low | Low | Low |
| Thomson et al., 1989 | Critical | Serious | Serious | Critical | Serious | Critical | Serious |
| Tlotleng et al., 2022 | Low | Prospective cohort study | Moderate | Low | Moderate | Low | Low |
| Wright et al., 2008 | Moderate | Low | Low | Moderate | Moderate | Low | Low |
| Wright et al., 2021 | Moderate | Low | Low | Moderate | Low | Low | Low |
| **ROB KEY:** | Low | | | | | | |
| | Moderate | | | | | | |
| | Serious | | | | | | |
| | Critical | | | | | | |
| | Prospective cohort study | | | | | | |

**Early childhood exposure (0–6 y).** Five of the included studies analyzed early childhood exposure (0 –≤6years), including Dietrich et al., [19] and Wright et al., [20, 21] using the Cincinnati cohort, Sampson and Winter using the Chicago cohort [25], Aizer and Currie using the Rhode Island cohort [29], and Emer et al., [31] using the Milwaukee cohort. The Cincinnati Lead Study reported no significant results when studying lead exposure from 3m – 6 y and parental or self-reports and delinquent behavior [19]. For average childhood lead levels (1m – 6 y), violent arrests and property arrests, no statistically significant results were reported by Wright et al., [20, 21].

Sampson and Winter (2018) did not find an association between lead exposure and arrests, but they did report an association between lead exposure and antisocial behavior, an outcome strongly associated with arrests (β for all arrests = 0.56 (95% CI [0.18–0.94]), violent crime arrests (β = 0.55, 95% CI [0.08–1.03]), and property crime arrests (β = 0.80, 95% CI [0.40–1.21]) [25]. Aizer and Currie reported a positive association between childhood lead levels and juvenile/adult detentions or incarcerations (β = 0.0014, p<0.001)) [29]. Emer et al., assessed average childhood lead levels and firearm violence perpetration and found that at blood lead levels between 5–10 μg/dl a RR of 2.5 (95% CI [1.5–4.1]) was reported, while at blood levels between 10–20 μg/dl a RR of 3.1 (95% CI [1.9–5.2]) was reported [31]. The Cincinnati Lead Study assessed exposure at the 5 to 6-year window of exposure (5–6 y). Positive associations were found when assessing 6-year blood lead levels and official data on adult arrests (IRR 1.07, 95 CI [1.00–1.14]), drug arrests (IRR 1.17, 95% CI [1.02–1.33]), and lifetime arrests (IRR 1.08, 95% CI [1.01–1.16]). This was also the case when assessing a range of average late childhood

(5–6.5y) lead levels and adult arrests (IRR 1.07, 95% CI [1.01–1.13]), drug arrests (IRR 1.13, 95% CI [1.03–1.24]), and lifetime arrests (IRR 1.06, 95% CI [1.00–1.13] [21]. It should be noted that despite a relatively small sample size (n = 254), most of the estimations maintain fairly narrow confidence intervals [20, 21].

**Late childhood exposure (6–11 y).** Late childhood lead exposure was assessed by 5 different studies in New Zealand, Edinburgh, Italy, and Pittsburgh. In New Zealand, Fergusson et al., reported an association between dentine lead levels and official conviction data (negative binomial coefficients for dentine lead levels 6–8 µg/g: 0.35, 9–11 µg/g: 0.52, and >12 µg/g 0.79 (p = 0.02); results for self-reported delinquency were not statistically significant) [28]. Beckley found an association between late childhood blood lead levels and criminal convictions (OR 1.23, 95% CI [1.00–1.51]), and nonviolent offenses (OR 1.28, 95% CI [1.01–1.61]), but no associations were found for one-time offenses, recidivistic offenders and violent offenders [27]. In the Edinburgh study, Thomson et al., reported a significant log-odds coefficient of 1.08 (p = .004), (i.e., the odds of being in a higher scoring category of poor behavior when blood lead levels increase by a factor of 2.72) [30]. In Italy, Renezetti et al., reported an association between lead exposure and both social problems and aggressiveness [34] (Table 3). Needleman made a similar assessment in Pittsburgh using reports by teachers and parents and reported an OR of 1.89 (95% CI [0.83–4.3]) for parental report of antisocial behavior and an OR of 2.16 (95% CI [0.96–4.6]) for teacher reports of antisocial behavior [26].

**Adolescence and adult exposure.** Five studies assessed the association between exposure in adolescence or adulthood and crime. Nkomo et al., assessed both violence and aggression in South Africa and found positive associations with increased blood lead levels at 13 years of age and physical violence at 15–16 years of age ($\beta$ = 0.26, p <0.01). When blood levels increased from <5µg/dl to >10 µg/dl, measures of direct aggression increased ($\beta$ = 0.34, p = 0.06). Direct aggression was defined by the authors as increased destruction of objects, attacking others, meanness, threatening to hurt others and physical altercations. No associations were found between lead exposure and indirect aggression (which included variables such as: having a hot temper, loudness, screaming, moodiness, argumentativeness, teasing others, and seeking attention), disobedience, violence using a weapon, fighting, sexual harassment, robbing or verbal and emotionally abusive behavior [22, 23]. Tlotleng et al., studied bone lead levels in young adults (23–24 y) and found that one microgram per gram increase in bone lead levels increased the mean aggressive score for anger by 0.25 (95% CI [0.04–0.37], scoring range 14–35). A similar increase was found for physical aggression, verbal aggression, and hostility, however these results did not reach statistical significance [24]. The Needleman et al., Pittsburgh-based case-control study reported a stronger relationship between bone lead levels and arrest or delinquent adjudication with an OR of 3.7 (95% CI [1.3–10.5]) [26]. Olympio et al., reported an OR of 3.04 (95% CI [1.07–8.64]) when assessing dental enamel lead levels and rule-breaking behavior in Brazil, but also reported no association with self-reported delinquency (OR = 0.93, 95% CI [0.40–2.20]) [33], while Naicker et al., (2018) demonstrated that gun range participants with BLL >10 µg/dL were significantly more likely to engage in hostile behavior than those with blood lead levels levels <10 µg/dL (OR 2.83, 95% CI [1.103–7.26]) [32].

## Discussion

### General interpretation

This systematic review identified a wide range of diverse outcomes between exposure to lead at multiple windows of development and later delinquent, criminal, and antisocial behavior. The range of outcomes that were significantly associated with lead exposure were primarily related to an arrest, incarceration, or conviction of some type (Wright et al., 2008; Wright et al., 2021;

Aizer and Currie, 2019; Dietrich et al., 2001; Fergusson et al., 2008; Beckley et al., 2018; Needleman et al., 2002), with increasing blood lead concentrations in childhood prospectively associated with later arrests and convictions in several studies [19–21, 25, 29, 31]. In addition to this association, 7 studies found strong associations between lead exposure and later delinquent or aggressive behavior irrespective of arrest status [22, 23, 25, 26, 30, 32, 33], with Sampson and Winter reporting "a plausibly causal effect of childhood lead exposure on adolescent delinquent behavior but no direct link to arrests [25]". Even in reviewed studies in which statistically significant associations between lead and crime did not exist, significant relationships between lead and damaging patterns of behavior that are more likely to lead to negative long-term outcomes were still present. Significant results were found even with very low concentrations of lead in blood, as exemplified by Renezetti et al., where mean blood lead levels were 0.94 µg/dL (SD 0.48) [34].

Despite these statistically significant associations, no clear findings regarding the association between lead exposure during specific developmental windows of exposure and the later development of criminal behavior emerged within the literature, with the available data pointing instead to an overall link between lead exposure and the later development of aggressive or hostile traits as well as criminal convictions and arrests.

## Trends in the evidence

A shift from lower to higher effect estimates as magnitude of lead exposure increased was a common and not unexpected outcome in our review. This finding is commensurate with existing ecological literature, although the magnitude of the effect size was much lower in our reviewed studies than that seen in ecological ones [7–9, 16–18]. Fergusson et al., demonstrated a dose-response relationship between adjusted mean numbers of violent convictions, property convictions, and self-reported offense and increasing blood lead levels [28]. Similarly, Emer et al., demonstrated an increase in the risk ratio (RR) from 2.5 (95% CI [1.5, 4.1]) at blood lead levels of 5 and <10 µg/dL to a RR of 3.5 (95% CI [2.1, 5.8]) at blood lead levels of 20 µg/dL [31]. Aizer and Currie report a similar dose response across their models that extended even to fully adjusted models that accounted for the interaction term between lead exposure and male sex, where every 1-unit increase in blood lead levels increased the probability of juvenile detention by 1.3 percentage points on a baseline rate of 1.8% [29].

Some studies demonstrated what might be described as conflicting results, with antisocial behavioral traits linked to lead exposure but not later criminal behavior or vice-versa [25, 26]. The Sampson and Winter study demonstrated that total arrests, violent crime arrests, property crime arrests, and all other types of arrest were not significantly associated with lead exposure. Despite this, the study authors demonstrated a significant association between childhood lead exposure and later antisocial behavior. The authors also demonstrated that antisocial behavior traits in wave 4 of their sample (ages 16–18) were significantly associated with all arrests and property crime arrests. As antisocial behavior traits are strongly associated with both criminal behavior and lead exposure, this association is noteworthy [25].

Of those studies assessing lead exposure and delinquent behavior using a form of self-report (either self-report, teacher-report, or parental-report), most demonstrated a small to moderate positive association between lead exposure and reports of delinquency and poor behavior. Thomson et al., reported a statistically significant log-odds coefficient of 1.08 (p = .004) when analyzing the association between aggressive/anti-social behavior and increases in log blood-lead [30]. Olympio et al., demonstrated that individuals with high levels of dentine lead (>217.35 ppm) were 2.87 times more likely to suffer from externalizing (antisocial) behaviors, 3.04 times more likely to experience social problems, and 3.72 times more likely to engage in

rule-breaking behavior than those exposed to low levels of dentine lead (<217.35 ppm) [33]. In the Nkomo et al., study, those with blood lead levels > = 10 μg/dL in early adolescence demonstrated a statistically significant increase in the risk of direct aggression [23]. Naicker et al., demonstrated that those gun range participants with blood lead levels greater than or equal to 10 μg/dL demonstrated 2.47 increased odds of hostile behavior than those with blood lead levels of less than 10 μg/dL [32]. Dietrich et al., demonstrated a significant association between all blood lead exposure variables (prenatal, 78 months, and average childhood) and self-reported delinquent behavior, and prenatal blood lead levels and parental-reported delinquent behavior [19]. All of these results point to a significant association between lead exposure and hostile, antisocial, and aggressive behavior- traits that strongly correlate with later criminal behavior [41, 42].

There are caveats to these findings and nuances within studies that should be discussed. Although Olympio et al., demonstrated positive associations between several negative behavior traits and dentine lead levels, the authors did not demonstrate any association between aggressive behavior or conduct problems and dentine lead levels. Similarly, the Nkomo et al., study reported that those with blood lead levels > = 10 μg/dL in early adolescence were not statistically significantly more likely to engage in behaviors involving indirect aggression [23]. While it is important to note this non-significant finding, it is also important to note that those children and adolescents who engage in behaviors of indirect aggression are less likely to demonstrate antisocial behavior traits than those children who engage in directly aggressive behavior [43]. As antisocial behavior traits are strongly linked to criminal behavior it is important to weigh findings of direct and indirect aggression accordingly [42, 43]. A close look at the R2 values within the Dietrich study indicate that the variance present was only moderately accounted for, a finding that is concerning for inadequate measure of confounding [19]. However, the authors address this and state this finding be due to the homogeneity seen across demographics rather than an inappropriate measure of confounding. This is supported by the excellent discussion on potential confounding variables outlined within the Dietrich et al., manuscript. Finally, Needleman et al., did not report a statistically significant odds ratio when assessing both parental and self-report of delinquent behavior, or teacher self-report, but did find a strong association between lead exposure and later arrests [26].

## Consistency with the existing literature

While our present study aims to build upon existing evidence to create a robust review of individual-level data, it is also worthwhile to consider how our review aligns with previous literature. Prior ecological studies based on aggregate evidence have strongly suggested that dramatic decreases in population lead levels (via removing lead from gasoline, banning leaded paint, etc.) were the impetus behind large decreases in violent and criminal behavior [7–9, 16]. The magnitude and significance of the findings of our review do not suggest that the striking results seen with the ecologic aggregated assessment translate to the individual level.

A team from the University of Glasgow conducted a meta-analysis on the association between lead exposure and crime [44]. Their study differed from ours in several key ways: 1) the Higney et al., study maintained a narrow definition of criminal behavior, including only those studies that explicitly specified crime or criminal behavior as an outcome, and excluding studies that had outcomes strongly related to crime, such as outcomes of aggression or delinquent behavior 2) the authors chose to only search the databases Web of Science, PubMed, and Google Scholar, 3) the authors chose to incorporate ecological as well as individual level data into their meta-analysis.

Although it is clear that there are catastrophic impacts of high lead exposure on the neurodevelopmental function of a child, low to moderate exposures still carry substantial risk. While

it has long been known that exposures to high levels of lead are associated with detrimental outcomes such as lowered intelligence quotient scores [45–47], lowered verbal, auditory, and speech processing scores [48], and poor focus and attention [49], it also been shown that exposure to low levels of lead in childhood predicts a lower intelligence quotient score [50, 51], conduct disorder [52], lower scores on tests of cognition [53], and poor neuromuscular development [54]. Low-level lead exposures may cause neurotoxic damage via interference with calcium ion channels [55]. As lead is structurally similar to calcium, it has been theorized that lead binds to the same channels within the brain that calcium would, thereby inhibiting neurotransmitter release and downstream functions such as cellular function and growth [55]. Additional theories suggest that lead may act as an N-Methyl-D-aspartate receptor (NMDA-R) antagonist, may act as a calmodulin agonist, may disrupt protein-kinase C function, and may directly damage mitochondria [55]. Compounding this is the unfortunate fact that lead exposure in the prenatal period has been demonstrated to induce negative long-term effects that appear to have "no evidence of a threshold" [56], a finding that should give us pause as we debate the value of mitigating low-dose exposures. Smaller exposures are still concerning, as small effect sizes can have a large impact at the population level, a point that has been emphasized by other authors and is particularly relevant to environmental exposures [38, 57]. From the perspective of economic cost, Aizer & Currie report that "exposure to even low levels of lead in early childhood generates substantial costs for many years after initial exposure" [29]. Those who suffer from low socioeconomic status, poor health outcomes, and racial inequities are also those most likely to experience the negative impacts of low-level, chronic lead exposures [58–60], and it is vital that we consider how these vulnerable populations may be impacted by low-level exposures.

## Augmentation of the ROBINS-E tool: A contribution

The "Risk of Bias in Non-randomized Studies of Interventions" (ROBINS-I) tool is a well-validated, standard tool that has been in use in clinical medicine since 2008 [61]. The ROBINS-I tools' foundational premise is that a randomized controlled trial remains the gold-standard in study design, thus all observational studies should be compared against a hypothetical randomized controlled trial design when assessing potential biases [61]. The ROBINS-E tool is based off the ROBINS-I tool, with a focus on exposure rather than intervention and with additional domains on outcome and exposure measurement included within the tool. The ROBINS-E tool can be accessed at https://www.riskofbias.info/welcome/robins-e-tool [15].

The ROBINS-E tool is a strong method for evaluating environmental exposures, however it is a time-consuming tool that assesses each domain of bias in considerable depth. For the purposes of our review, we chose to use an abbreviated and augmented version of the ROBINS-E based on the needs of our study. We chose to add a separate risk of bias table for case-control and cross-sectional studies, as the ROBINS-E tool is designed primarily for cohort studies, however it should be noted that we leaned very heavily on the ROBINS-E design and recommendations throughout this process.

**Augmentation of the ROBINS-E tool: Included confounders.** Consideration and inclusion of appropriate confounding variables is integral to the process of demonstrating an association between lead exposure and crime. When considering what confounders would be integral to a low risk-of-bias classification in our review, we assessed the literature to see which factors would have the greatest impact on confounding and attempted to minimize those factors to include only those variables that would be applicable to the studies chosen for our review. We determined that any study that failed to consider potential confounders would automatically be ranked as "very high risk of bias" in the domain of confounding (S3 Table).

It has been well documented that children from economically disadvantaged households tend to exhibit heightened susceptibility to the various detrimental effects caused by lead exposure [3, 62, 63], and are simultaneously more likely to be born into homes where higher exposures of environmental lead contamination are present [5]. For these reasons, we considered socioeconomic status, or a proxy thereof, to be an important potential confounder in our review.

Two authors did not define how they measured socioeconomic status but clearly stated that they had included the variable as a confounder in their analyses [27, 28]. Several studies alluded to the inclusion of socioeconomic status by listing confounding variables that typically act as proxies for measurement, but never clearly stated which variable, or combination of variables, acted as the proxy [21, 26, 30, 32, 33].

Another confounder of concern is that of the home environment. While it is difficult to parse out what exactly makes a home environment one in which a child can thrive, the literature suggests that higher maternal IQ and/or higher Home Observation for Measurement of the Environment (HOME) scores are associated with a reduction in the negative impacts of lead exposure and improvements in cognitive developments [64, 65]. Maternal socioeconomic status may play into the structure of the home environment by acting as an important predictor of maternal blood lead levels [65]. As we know that increased maternal blood lead is strongly linked to decreased IQ and increased levels of lead exposure in both the mother and the infant [66–68], and that socioeconomic status is a strong predictor of lead exposure at baseline [3, 64, 65, 69], we felt that including maternal socioeconomic status as a proxy for home environment would be acceptable, particularly in combination with other variables such as maternal IQ or HOME scores.

The final key confounder we identified was race and/or ethnicity. African American children have, on average, higher mean blood lead levels compared to other races [70, 71]. In the United States, individuals of color are also more likely to be arrested or incarcerated for criminal behavior [72–74], we decided that race/ethnicity would be our final required confounder for a low-risk-of-bias for confounding designation for studies where race is not uniformly distributed within their countries or cohorts. We did not require race to be a confounder for studies that sampled from a homogenous racial distribution (S3 Table).

Although there are many other confounders that could have been potentially included, the confounders listed above have been demonstrated by the current literature to be the most integral to an accurate assessment of the relationship between lead and crime. Many studies chose to include age and gender as confounders or chose to stratify by age and gender. The degree to which lead negatively impacts neurodevelopment is impacted by a child's age, the degree to which the child was exposed to lead, and the amount of lead the child was exposed to [52, 75]. Age is thus a multi-faceted variable that may act as a potential confounder, an interaction term, or a variable by which to stratify results, depending on the study design, thus we decided not to downgrade a study if age was not used as a confounder. Similarly, the sex of the participant was another problematic variable to consider. While criminal behavior is overwhelmingly seen more frequently in men versus women [76], it is not agreed that boys consistently demonstrate higher blood lead levels than girls. There are sex-specific trends in regions known to have higher baseline levels of environmental contamination [77, 78], but no widespread agreement that these trends are universal. Therefore, we again decided to leave the designation of sex as a potential confounder, an interaction term, or a variable by which to stratify results to the study authors.

While most of the studies were within the "Low" or "Moderate" risk of bias categories, two studies were part of the "Serious" risk category: Naicker et al., and Thomson et al.,. Both

Thomson et al., and Naicker et al., demonstrated statistically significant results. Despite this, excluding the two studies listed above would not result in a different overall conclusion.

## Considerations for future research

**Mediation and moderation.** In 11 of the 17 total studies (excluding Sampson & Winter, 2018; Thomson et al., 1989; Wright et al., 2021; Wright et al., 2008; Aizer & Currie, 2019; Tlotleng et al., 2022) potential mediators, moderators, and interaction terms were either not included or not discussed as part of the statistical analysis process. This may have been because the study authors felt confident that their covariates acted as confounders only and not as mediators in the causal pathway, or it may have been because the relationship between lead and crime is so complex that accurately parsing out mediating relationships was viewed as a largely impractical task. It has been previously hypothesized that the lead-crime relationship could be mediated by low intelligence quotient (IQ) scores in the child, attention-deficit-hyperactivity disorder (ADHD), poor school performance, and/or the presence of an abnormally high number of delinquent peers [79]. The magnitude to which these variables impact the causal relationship, and the reliability and validity with which these variables are measured, remain a source of debate and quantifying these interactions is a difficult task. Additionally, many journals either do not require or do not grant adequate space for mediation analysis in observational data [80]. Despite this, failing to account for mediators can bias the effect estimate, and as such, future studies should make every effort to account for these potential relationships whenever possible [81–86].

## Limitations of the evidence

There was marked heterogeneity in the outcome measures assessed across studies. For example, Wright et al., reported on the outcomes of adult arrests, violent arrests, drug arrests, property arrests, and lifetime arrests [21], while Nkomo et al., reported on the outcomes of violence using a weapon, physical violence, fighting, sexual harassment, robbing, and verbal/emotionally abusive behavior [22]. It is likely that there was substantial overlap in the categorization of various criminal behaviors across studies but delineating the exact boundaries of these categories was challenging. This heterogeneity made it impossible to complete a meta-analysis on individual-level markers of exposure in the setting of our criminal behavior outcome.

A second limitation of the evidence was found in the heterogeneity demonstrated across statistical reporting modalities. A combination of incidence rate ratios, odds ratios, means, and principal component analysis scores were reported. Comparing across all outcomes allowed us to gain a general picture of the overall evidence but assessing the precise magnitude of the risk across studies was difficult due to the variance in outcomes assessed.

A third limitation was how authors studied windows of exposure. Two critical windows of exposure have been identified in the literature; a prenatal exposure window and an early childhood exposure, measured around 6 years of age [54, 87]. Although the available studies for these windows showed significant results (Dietrich and Wright), all participants came from the same cohort. Further research is needed to increase our understanding surrounding critical windows of exposure to lead and criminal behavior.

A final limitation was found in the limited generalizability. The participants in the Olympio et al., study were located in the slums of Ferradura Mirim, Brazil [33], while Beckley et al., sampled from the full range of available socioeconomic classes in the South Island of New Zealand [27], and Wright et al., drew from the poorest regions of Cincinnati, Ohio (USA) [20, 21]. While the diversity in countries and socioeconomic classes studied yielded a rich and valuable addition to the knowledge surrounding lead exposure outcomes, that same variability made it

difficult to apply the results of these studies to any one population. Additionally, there was no data available from within Asia, a fact which creates an obvious gap in our current knowledge base about individual level lead exposures and crime. As data from the United States was over-represented in this study, we strongly recommend that researchers consider focusing on more diverse populations in future analyses.

## Conclusion

Children do not absorb or metabolize lead in the same way as adults and are far more suscepti-ble to the negative impacts of lead exposure due to a hyper-permeable blood-brain barrier and rapidly developing organ systems [88–90]. Animal studies have demonstrated adverse neuro-behavioral effects in animals exposed to lead [10, 11], and multiple ecological studies have demonstrated an association between lead exposure and criminal behavior [7–9, 16]. This review demonstrates an association between exposure to lead and the later development of delinquent, antisocial, and criminal behavior. Although borderline levels of risk are seen in several of our included studies, most are above the null value and estimates of risk are generally precise. While the magnitude of the risk varied depending upon the outcome assessed and the adequacy of the confounders included, most studies employed robust measures of exposure and outcome assessment. Criminal behavior exists on a broad spectrum, and each study chose to delineate the limits of that spectrum in a different way. We propose that future studies should be carried out in a more diverse range of countries and focus on adequate assessment and control of relevant confounders and utilize a common set of indicators for both exposure and outcome in order to measure the overall impact of lead through a quantitative meta-analy-sis. There is a paucity of original data at the individual level on the effects of lead exposure in childhood and later criminal behavior and more evidence is necessary to evaluate the strength of the associations seen in this review. Despite these limitations, this review in conjunction with the available biological evidence demonstrates that an excess risk for criminal behavior in adulthood exists when an individual is exposed to lead in utero or within childhood.

## Supporting information

**S1 Text. Search strategy.**
(DOCX)

**S1 Table. Description of the adapted ROBINS-E framework for assessing risk of bias in environmental health studies for prospective and retrospective cohort studies.**
(DOCX)

**S2 Table. Description of the adapted ROBINS-E framework for assessing risk of bias in environmental health studies for cross-sectional and case-control studies.**
(DOCX)

**S3 Table. Justification for risk of bias designations.**
(DOCX)

**S4 Table. Variation in confounders by study author.**
(DOCX)

## Author Contributions

**Conceptualization:** Maria Jose Talayero, Carlos Santos-Burgoa.

**Investigation:** Maria Jose Talayero, C. Rebecca Robbins.

**Methodology:** Maria Jose Talayero, C. Rebecca Robbins.

**Project administration:** Maria Jose Talayero.

**Supervision:** Emily R. Smith, Carlos Santos-Burgoa.

**Validation:** Maria Jose Talayero, C. Rebecca Robbins, Emily R. Smith, Carlos Santos-Burgoa.

**Writing – original draft:** Maria Jose Talayero, C. Rebecca Robbins.

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
