## [Decision Letter · Decision Letter 0]

3 Apr 2023

PGPH-D-22-02052

The association between lead exposure and crime: a systematic review

Dear Dr. Maria Jose Talayero Schettino,

Thank you for submitting your manuscript to PLOS Global Public Health. After careful consideration, we feel that it has merit but does not fully meet PLOS Global Public Health’s publication criteria as it currently stands. Therefore, we invite you to submit a revised version of the manuscript that addresses the points raised during the review process.

While the research topic is interesting and relevant, there are major issues that should be taken into consideration before the article is accepted for publication. The writing and structure of the manuscript needs a lot of improvement. The authors need to go through the manuscript in great detail and address the reviewers' comments.

We look forward to receiving your revised manuscript.

Kind regards,

Naveen Puttaswamy, Ph.D

Academic Editor

Journal Requirements:

1. We have noticed that you have uploaded Supporting Information files, but you have not included a list of legends. Please add a full list of legends for your Supporting Information files after the references list.

Additional Editor Comments (if provided):

Reviewers' comments:

Reviewer's Responses to Questions

**Comments to the Author**

1. Does this manuscript meet PLOS Global Public Health’s publication criteria? Is the manuscript technically sound, and do the data support the conclusions? The manuscript must describe methodologically and ethically rigorous research with conclusions that are appropriately drawn based on the data presented.

Reviewer #1: Yes

Reviewer #2: Yes

Reviewer #3: Yes

Reviewer #4: Partly

Reviewer #5: Partly

2. Has the statistical analysis been performed appropriately and rigorously?

Reviewer #1: Yes

Reviewer #2: Yes

Reviewer #3: I don't know

Reviewer #4: N/A

Reviewer #5: N/A

3. Have the authors made all data underlying the findings in their manuscript fully available (please refer to the Data Availability Statement at the start of the manuscript PDF file)?

Reviewer #1: Yes

Reviewer #2: Yes

Reviewer #3: Yes

Reviewer #4: Yes

Reviewer #5: Yes

4. Is the manuscript presented in an intelligible fashion and written in standard English?

Reviewer #1: Yes

Reviewer #2: Yes

Reviewer #3: Yes

Reviewer #4: Yes

Reviewer #5: Yes

5. Review Comments to the Author

Reviewer #1: The association between lead exposure and crime is a complex issue that has been the subject of much research and discussion over the years.

The study has utilized the best methodology for its objective, and it is well described. It is not clear why the authors emphasized the elimination of six major ecological studies. It does not affect the conclusions of the article, but it would be interesting to state the reasoning behind this decision, since it was highlighted in the description of the studies that were included or not included.

The discussion of confounding variables is of particular relevance to this article and was well done. Mentioning it in the abstract could be beneficial for the reader.

The conclusions and suggestions for further research are also very important and are clearly stated.

Reviewer #2: Minor comments

• This scientific work assessed the association between lead exposure and crime, my suggestion to the authors is to explain molecular mechanisms of the potentially hazardous effects of lead exposure.

• Figure 1: The authors mentioned that the total studies excluded were 50, however, the summation of these studies is 49 (34+10+3+2). Please, correct that.

• In lines 56 and 57, the authors raise a concern about lead exposure without mentioning the sources of exposure, can the authors give some details about the sources of exposure in developing and developed countries?

• In line 65, the authors mentioned inconsistent results of the epidemiological studies and on page 28, some conflicting data regarding antisocial behaviour, can the authors add some more explanation to these inconsistencies?

• Some technical problems occurred many times in the text, “Error! Bookmark not defined”. Can the authors resolve this technical issue?

• The authors mentioned on page 35 that the explanation of the mechanisms of the potentially hazardous effects of low-level lead exposure are not fully understood. Can the authors add some of the published mechanisms?

• In the last sections the authors mentioned the confounders that may impact the association with criminal behaviours. Can the authors shed the light besides the socioeconomic status, the nutritional status of both mother, infants, and children and exposure to other environmental contaminants like other heavy metals and or pesticides which may impact this association?

• The authors mentioned in table 2 the location where the previous studies were conducted and according to this table, many of these studies were conducted in the USA and no study was carried out in Asia. So, in the recommendation section, can the authors add a recommendation of conducting studies in other locations to better assess this relationship by putting into consideration gene-environments interaction and the difference in the social and economic determinates?

Reviewer #3: A very nice analysis of available studies from Europe, Americas, South Africa and Oceana.

No asian study included in the analysis, any particular reson for that?

Colclusion is made in line with the results of the included studies. But I feel further diversication on available studies could have given a better global persective.

Reviewer #4: The authors sought to conduct a systematic review of studies with individual-level measures of endogenous lead and crime, broadly defined. This is an important feature as ecological studies may be strongly confounded by contextual effects such as neighborhood characteristics. The methods, design, and summary of findings are well-articulated and particular strengths include the librarian-guided search and application of relevant guidelines, particularly ROBINS-E.

That being said, more could have been done to contextualize the state of the literature, particularly in relation to the recent meta-analyses by Higney, et al. they briefly mention. The previous study, conducted by (I believe) economists, give ample room for the current authors to do a more in depth evaluation of the individual-level studies identified here. For example, Higney, et al point out (it is mentioned here, too) that ecological studies showed substantially larger effect sizes than individual-level studies, with one possibility being, e.g. neighborhood confounding. Was there a discussion or assessment of neighborhood factors in the current set of studies? I was surprised that contextual factors were not even mentioned in the list of critical confounders. Further, Higney, et al identified substantial publication bias. Was this evaluated here? Did the studies with smaller / negative effect sizes have stronger designs / less risk of bias? On that subject, given the use of ROBINS-E, I had hoped there would be greater discussion & comparisons regarding key features of the included studies to draw out patterns and make recommendations for future studies. Are certain exposure measurements and study designs more credible than others as experimental analogues? Which design/analytic features were related to smaller or larger effect sizes (or null findings). The use of nominal statistical significance (i.e. p-value thresholds) to compare findings, such as across ages of exposure, does not seem particularly relevant here, especially as sample sizes varied greatly. Standardized effect sizes would be much more useful (along with the estimates of variability). On that note, it was difficult to understand the various effect sizes presented as they were unit-less (how much of an increment in lead?) and varied between betas and IRRs. More stratification and comparisons of strength of evidence by risk of bias would be welcome.

Finally, the authors rightfully mentioned that understanding mediators and moderators would strengthen the evidence for a causal link between lead and crime, particularly with respect to alternatives hypotheses regarding education, IQ, aggressive behaviors, opportunity, etc. Was there anything in the studies reviewed that specifically support or point in these directions? Do differences in effects across types of outcomes, e.g. violent vs. non-violent, point in a given direction? What about support for the hypothesis that age 6 is a critical window? What specifically should be looked at next to better understand these factors?

Overall, while the authors were clear in laying out the variety of settings for the studies reviewed, it was hard to draw firm conclusions about the available evidence, and the conclusion pointing strongly to effects on crime seem overstated particularly in light of the issues raised for Higney, et al's meta analysis. Given the flexibility afforded by their choice to do a narrative review (rather than meta-analyses), it would seem an ideal opportunity for a deeper assessment of the state of research and provide concrete steps forward.

Other points:

-Table 4: Thomson et al is 1989 not 1996?

-In the Results, it would be helpful to see the units of each of the measures even if they cannot be harmonized

-Discussion, General Interpretation: "consistent with epidemiological research..." aren't all these studies reviewed epidemiologic research? Perhaps this should be rephrased to specifically discuss the critical period / life course models presented in the given citations (39,40)

-Give more context for the scientific plausibility behind age 6 as a critical period (and present any relevant findings from the included studies)

-Trends in Evidence: Again it would be helpful to see a breakdown in the effect sizes / presence / absence of findings based on risk of bias or other design features -- Did any adjust for neighborhood characteristics? did those have smaller effect sizes?

-"the form of sampling makes the authors finding of an increase odds...even more remarkable": Please clarify the reasoning behind why this is remarkable. Selection bias on the basis of self-selection into participation can drive effect estimates in any direction based on what reasons people choose to participate. For example, those from poorer neighborhoods (with higher lead) and more antisocial behavior may choose to participate leading to this observation.

-"the authors [Dietrich study] address this and state this finding [may] be due to..." not sure how this statement is supported. Understanding this is a paraphrase of the original paper, do the current authors think it's justified?

-"the authors chose to incorporate ecological as well as individual level data...may obscure the true impact of lead exposure": How would it obscure if effect sizes of ecological studies are orders of magnitude higher? Does this sentence just mean the estimates are biased? I believe Higney, et al make it clear that those effects are likely biased and give a much smaller estimate of effect in their meta-analysis. Can you clarify this statement, otherwise it appears you are misrepresenting those authors' findings.

Reviewer #5: This paper makes a strong effort at addressing a critical gap identified by the authors – the lack of individual-level data regarding lead exposure and crime, which has led to a frequent reliance on ecological studies. It is clear that the authors have put substantial work into distilling the identified studies into a single narrative. As the authors note, the possibility to draw the results of these studies together into a larger analysis is limited by the variety in reported exposure measures, outcomes, and summary statistics.

Major comments:

1. A topic as sensitive as this one requires a great deal of care and precision with language. The primary research question emphasizes “crime” but then a much wider array of outcomes is included and discussed (e.g. “aggression”, “anti-social behavior”, “rule-breaking behavior”). Furthermore, some of these terms are not defined, for example “indirect aggression.” I would recommend reviewing the language used to ensure it is consistent and accurately reflects what is ultimately emphasized in the manuscript.

This recent paper may provide some additional insight: Rachel M. Shaffer, Jenna E. Forsyth, Greg Ferraro, Christine Till, Laura M. Carlson, Kirstin Hester, Amanda Haddock, Jenna Strawbridge, Charles C. Lanfear, Howard Hu, Ellen Kirrane. Lead exposure and antisocial behavior: A systematic review protocol. Environment International, Volume 168, 2022, 107438, ISSN 0160-4120, https://doi.org/10.1016/j.envint.2022.107438.

2. While it is encouraging that there were eligible studies from Brazil and South Africa, a slightly expanded discussion on the geographic limitations of this review is warranted. This could include the lack of any low-income countries, no representation from Asia, and importantly, an overrepresentation from the US (8 of the 17 studies were from the US, and 3 from just one city).

3. It appears that there is an overlap in the categories for “age at exposure measure” (<6, 5-6, 6-11). This becomes particularly confusing in the “General Interpretation” section, where the prenatal and 5-6 year period are highlighted as being of particular concern regarding negative outcomes.

I have some doubts regarding this statement - “This review allows us to clearly define a window of time in which exposure is linked to a higher risk of later criminal behavior.” There does not appear to be sufficient data to narrow a window of concern so precisely. This conclusion appears to be based on just the Cincinnati cohort (N=254), and as the authors note themselves, quite narrow confidence intervals.

For the broader 0-6yr category, the results are mixed. Wright et al 2008 and 2001 and Dietrich et al 2001 include children up to 6 years, and none of the associations reported in Table 3 under that age group were significant. The direct associations between lead and negative outcomes for that age were not significant in Sampson & Winter either.

4. In the bias assessment, it is troubling that 7 of the 17 studies have serious or critical concerns about confounding. As the authors note, factors like socioeconomic status and race are hugely influential on lead exposure levels, and independently, on outcome variables like arrest rates. The authors could examine how excluding the studies found to have a high-level of bias (either for particular bias categories or the overall bias metric) influences the conclusions.

5. To the reader, there is inconsistency in how the authors present the strength of the conclusions which can be drawn from the results of this systematic review. In the abstract, the authors highlight the “consistent and statistically significant” association seen in prenatal and 5-6 y period. However, this narrows the dataset to a single, relatively small cohort (Cincinnati) and draws the focus away from the very mixed results of the overall systematic review. I think the results are represented more faithfully later in the manuscript - "Some studies demonstrated what might be described as conflicting results, with antisocial behavioral traits linked to lead exposure but not later criminal behavior or vice-versa" and “The magnitude and significance of the findings of our review do not suggest that the striking results seen with the ecologic aggregated assessment translate to the individual level."

Minor:

1. There are some inconsistencies in the reporting of the numbers of studies. For example, in the abstract, it is reported that there are 14 BLL studies, 3 bone lead, 1 dentine, but these numbers are different in Table 2. In line 201, it states that 9 studies are from the US but Table 2 indicates 8.

2. I would recommend some acknowledgement of the fact that bone and dentine lead reflect cumulative exposure, while blood lead provides a snapshot of recent exposure.

3. Line 45. "individuals" rather than "communities" may be more appropriate here, as the references cited do not appear to reflect population level trends in these health outcomes.

4. Line 292. Refers to "African Americans" but one of these studies is from South Africa.

5. Section "Adolescence and adult exposure" (no page numbers) - typo "studid"

6. Section "Trends in evidence" - 1st sentence. I believe this refers to the severity or magnitude of lead exposure, not "rates"

7. Section "Augmentation of the ROBINS-E Tool: Included Confounders" - African Americans are referenced explicitly again here, but as this paper is international in scope, are there other relevant trends regarding lead exposure and race identified in other countries?

6. PLOS authors have the option to publish the peer review history of their article (what does this mean?). If published, this will include your full peer review and any attached files.

**Do you want your identity to be public for this peer review?** For information about this choice, including consent withdrawal, please see our Privacy Policy.

Reviewer #1: No

Reviewer #2: No

Reviewer #3: No

Reviewer #4: **Yes: **Jonathan Y Huang

Reviewer #5: No

---

## [Decision Letter · Decision Letter 1]

29 May 2023

PGPH-D-22-02052R1

The association between lead exposure and crime: a systematic review

Dear Dr. Schettino, 

Thank you for submitting your manuscript to PLOS Global Public Health. After careful consideration, we feel that it has merit but does not fully meet PLOS Global Public Health’s publication criteria as it currently stands. Therefore, we invite you to submit a revised version of the manuscript that addresses the points raised during the review process.

Thank you for submitting the revisions. The reviewers are satisfied with the revision and appreciate the effort of the authors to improve the manuscript. However, one of the reviewers has minor comments. Please respond with your revision at the earliest.

We look forward to receiving your revised manuscript.

Kind regards,

Naveen Puttaswamy, Ph.D

Academic Editor

Journal Requirements:

Additional Editor Comments (if provided):

Reviewers' comments:

Reviewer's Responses to Questions

**Comments to the Author**

1. If the authors have adequately addressed your comments raised in a previous round of review and you feel that this manuscript is now acceptable for publication, you may indicate that here to bypass the “Comments to the Author” section, enter your conflict of interest statement in the “Confidential to Editor” section, and submit your "Accept" recommendation.

Reviewer #1: All comments have been addressed

Reviewer #5: (No Response)

2. Does this manuscript meet PLOS Global Public Health’s publication criteria? Is the manuscript technically sound, and do the data support the conclusions? The manuscript must describe methodologically and ethically rigorous research with conclusions that are appropriately drawn based on the data presented.

Reviewer #1: Yes

Reviewer #5: Yes

3. Has the statistical analysis been performed appropriately and rigorously?

Reviewer #1: Yes

Reviewer #5: N/A

4. Have the authors made all data underlying the findings in their manuscript fully available (please refer to the Data Availability Statement at the start of the manuscript PDF file)?

Reviewer #1: Yes

Reviewer #5: Yes

5. Is the manuscript presented in an intelligible fashion and written in standard English?

Reviewer #1: Yes

Reviewer #5: Yes

6. Review Comments to the Author

Reviewer #1: No further comments

Reviewer #5: Thank you for the opportunity to review the revised manuscript. The authors have clearly put significant effort into the revision. I feel the discussion section of the current version more accurately reflects the aggregated results and the limitations of the existing data. In particular, the authors addressed my concerns about the windows of sensitivity for lead exposure; they expanded the section on the generalizability of the findings in terms of geography; and they made the statements regarding the strength of the study’s conclusions more consistent.

Several comments related to my previous feedback:

Line 557: I would suggest you indicate that this is specific to the US, or else include non-US references. Does this trend hold true in other countries (South Africa?)

559: How was it determined whether the studies sampled from a homogenous racial distribution? Was race reported for all included studies (but not necessarily included as a confounder)?

Some minor comments from a second read-through:

372: If you report the numeric change in aggression score, perhaps include the scale of the test for context.

379: This is the first time shooters is mentioned – explained later as “gun range participants” (440)

434: I would recommend rewording this as “individuals with high levels of dentine lead” (the dentine lead is the result of exposure, not the source of it)

440+442: Microgram/dL (not milligram). Also, sometimes mcg/dL is used and sometimes µg/dL throughout manuscript

536: What does the word “sensitive” here mean? That the negative effects of a comparable lead exposure level are more pronounced in a lower SES child than a higher SES child?

547: Please define/explain HOME acronym

561: While I completely agree that cultural factors such as use of lead-glazed ceramics would impact the magnitude of lead exposure in an individual, it is not clear to me how such factors would influence the relationship between lead exposure and crime/antisocial behavior.

597: What is implied by “unconventional parenting”? This seems vague and very subjective.

7. PLOS authors have the option to publish the peer review history of their article (what does this mean?). If published, this will include your full peer review and any attached files.

**Do you want your identity to be public for this peer review?** For information about this choice, including consent withdrawal, please see our Privacy Policy.

Reviewer #1: No

Reviewer #5: No

---

## [Decision Letter · Decision Letter 2]

23 Jun 2023

The association between lead exposure and crime: a systematic review

PGPH-D-22-02052R2

Dear Dr. Maria Jose Talayero Schettino,

We are pleased to inform you that your manuscript 'The association between lead exposure and crime: a systematic review' has been provisionally accepted for publication in PLOS Global Public Health.

Best regards,

Naveen Puttaswamy, Ph.D

Academic Editor

Reviewer Comments (if any, and for reference):

Reviewer's Responses to Questions

**Comments to the Author**

1. If the authors have adequately addressed your comments raised in a previous round of review and you feel that this manuscript is now acceptable for publication, you may indicate that here to bypass the “Comments to the Author” section, enter your conflict of interest statement in the “Confidential to Editor” section, and submit your "Accept" recommendation.

Reviewer #5: All comments have been addressed

2. Does this manuscript meet PLOS Global Public Health’s publication criteria? Is the manuscript technically sound, and do the data support the conclusions? The manuscript must describe methodologically and ethically rigorous research with conclusions that are appropriately drawn based on the data presented.

Reviewer #5: Yes

3. Has the statistical analysis been performed appropriately and rigorously?

Reviewer #5: N/A

4. Have the authors made all data underlying the findings in their manuscript fully available (please refer to the Data Availability Statement at the start of the manuscript PDF file)?

Reviewer #5: Yes

5. Is the manuscript presented in an intelligible fashion and written in standard English?

Reviewer #5: Yes

6. Review Comments to the Author

Reviewer #5: Thank you for addressing my previous comments. No further recommendations from my side.

7. PLOS authors have the option to publish the peer review history of their article (what does this mean?). If published, this will include your full peer review and any attached files.

**Do you want your identity to be public for this peer review?** For information about this choice, including consent withdrawal, please see our Privacy Policy.

Reviewer #5: No
